# Performing under Pressure: Insights into the Diagnostic Testing Burden at a UK National Health Service Clinical Virology Laboratory during the SARS-CoV-2 Pandemic

**DOI:** 10.3390/v14102233

**Published:** 2022-10-12

**Authors:** Paul William Bird, Georgina Taylor, Jessica Cafferata, Judi Gardener, Claire L. McMurray, Oliver Fletcher, Oliver T. R. Toovey, Christopher W. Holmes, Julian W. Tang

**Affiliations:** 1Department of Clinical Microbiology, University Hospitals of Leicester NHS Trust, Leicester LE1 5WW, UK; 2Department of Respiratory Sciences, University of Leicester, Leicester LE1 7RH, UK

**Keywords:** COVID-19, SARS-CoV-2, pandemic, diagnostics, non-SARS-CoV-2, routine, testing, burden

## Abstract

UK National Health Service (NHS) Clinical Virology Departments provide a repertoire of tests on clinical samples to detect the presence of viral genomic material or host immune responses to viral infection. In December 2019, a novel coronavirus (SARS-CoV-2) emerged which quickly developed into a global pandemic; NHS laboratories responded rapidly to upscale their testing capabilities. To date, there is little information on the impact of increased SARS-CoV-2 screening on non-SARS-CoV-2 testing within NHS laboratories. This report details the virology test requests received by the Leicester-based NHS Virology laboratory from January 2018 to May 2022. Data show that in spite of a dramatic increase in screening, along with multiple logistic and staffing issues, the Leicester Virology Department was mostly able to maintain the same level of service for non-respiratory virus testing while meeting the new increase in SARS-CoV-2 testing.

## 1. Introduction

National Health Service (NHS) Pathology Departments in the United Kingdom (UK) consist of several main specialities: Clinical Biochemistry, Haematology, Histopathology and Cytology, Clinical Genetics and Cytogenetics, Immunology, and Microbiology and Virology [1]. An estimated 70–80% of all healthcare decisions are based on the diagnostic results of analysing clinical patient specimens by Pathology Departments [2]. The primary role of a Clinical Virology Department is to provide a repertoire of tests on clinical specimens to detect the presence of viral antigens or antibodies by serological tests and/or detect viral DNA/RNA by molecular tests (e.g., the polymerase chain reaction, PCR).

On 31 January 2020, the first cases of severe acute respiratory syndrome coronavirus 2 (SARS-CoV-2) were confirmed in the UK and NHS laboratories across the country prepared to screen for the novel virus [3]. The scale of testing required was enormous in comparison to historic levels of testing for respiratory viruses, and the Government provided additional funding for consumables, assays and staff to meet demand. While screening of patients was left to the hospital laboratories (Pillar 1 testing) [4], the Government established so-called ‘lighthouse’ laboratories (based outside of the NHS network) for community screening (Pillar 2 testing) to provide high-throughput screening and testing capacity whilst minimising the impact on hospital labs [5].

As the pandemic progressed there were multiple waves of infection caused by mutations in the SARS-CoV-2 genome, creating new viral variants [6]. As these waves caused spikes in infections, hospital admissions fluctuated, and the Government placed the country in a series of national and regional lockdowns. The lockdowns restricted movement unless people were making necessary trips (e.g., food shopping) or were essential workers (NHS staff, police officers, fire services, etc.).

By June 2020, the Government announced that country-wide restrictions would be relaxed, with the exception of Leicester which was the first region to be subjected to ‘local lockdown’ due to high cases of SARS-CoV-2 [7,8,9]. In the subsequent months Leicester would be repeatedly place in and out of lockdowns as new variants emerged but testing demand and pressure on the Virology Departments did and has not decreased.

Over the course of the pandemic, NHS laboratories strived to meet demands of PCR SARS-CoV-2 screening. In September 2021, the Department of Health and Social Care (DHSC) released a document outlining the direct and indirect impact SARS-CoV-2 had on the health of the population. The paper reported that the diagnosis of a range of chronic conditions decreased significantly, as did the diagnosis of non-SARS-CoV-2 infections [10]. Yet, limited data has been provided on the ramifications SARS-CoV-2 has had on the other routine diagnostic serological and molecular services provided by Virology Departments.

Here, we present data on the rates of routine diagnostic viral tests performed before, and during the pandemic, to investigate what effects SARS-CoV-2 has had on an NHS Virology Department and its ability to perform non-SARS-CoV-2 services. Serology testing was mostly unaffected during the pandemic, so this study reports the impact of the pandemic on PCR testing only.

## 2. Methods

### 2.1. Data Collection

An electronic search of University Hospitals of Leicester (UHL) laboratory information management system was performed to collate all molecular and serological results for all virology samples between January 2018 to May 2022, screening for: Human Immunodeficiency Virus (HIV), Hepatitis B Virus (HBV), Hepatitis C Virus (HCV)—using the HIV, HBV, HCV Quant Dx Aptima Kits on the Panther platform (Hologic Inc., Manchester, UK); Cytomegalovirus (CMV), Epstein–Barr Virus (EBV)—using the Artus PCR kits (QIAGEN, Hilden, Germany); Herpes Simplex Virus (HSV), Respiratory Viruses(RV [IVA, IVB, RSV]—using the PCR kits from AusDiagnostics UK Ltd. (Chesham, Bucks, UK); Norovirus, Rotavirus, Adenovirus—using the RIDA^®^QUICK norovirus (R-Biopharm AG, Darmstadt, Germany) and ProFlow Adenovirus-Rotavirus Dual Test (Pro-Lab Diagnostics, Merseyside, UK) kits; and SARS-CoV-2—using a variety of kits and platforms including lab-developed tests using the initial WHO assay [11], the AusDiagnostics assays (Coronavirus Typing (8-well) assay, SARS-CoV-2, Influenza and RSV (8-well) assay, and Respiratory Virus (16-well) assay) [12], Aptima SARS-CoV-2 TMA assay [13], Cepheid Xpert Xpress Assay (Cepheid UK Ltd., High Wycombe, UK) [14], and Amplidiag and Novodiag SARS-CoV-2 and RESP-4 kits (Mobidiag UK, Slough, UK) [15].

### 2.2. Data Analysis

The data were sub-divided by viruses and de-duplicated. To give an accurate representation of tests being performed (requiring laboratory staff to book in, process and test individual samples), several results were merged and specific assays tested for several viruses during one test. All results were used to give a representation of the workload experienced within the laboratory before and after the peak of the SARS-CoV-2 pandemic. The data were analysed to calculate: the number of samples to detect/not detect viral DNA/RNA or presence of immunological response, the total tests performed, and the percentage of samples that were positive and negative during the time period.

All analyses were performed using Microsoft Excel (Microsoft Office Professional Plus 2010, Redmond, WA, USA). Samples were collated into specific groups: Blood Borne Viruses (BBV [HIV, HBV, HCV]), Respiratory viruses (RV [IVA, IVB, RSV]), Herpes Viruses (HV [HSV, CMV, EBV]), Diarrhoeal Viruses (DV [Norovirus, Rotavirus, Adenovirus]) and SARS-CoV-2.

### 2.3. Ethics

The data were acquired and analysed as part of a service evaluation of the routine diagnostic laboratory virology service provisions during the pandemic, no patient identifiable data (name, sex, date of birth, ethnicity) were collected or included in the study.

## 3. Results

The data show that pre-SARS-CoV-2 pandemic in 2018 and 2019 the virology laboratory processed 113,067 and 130,634 tests per year, respectively (2018: 25,729 RV, 844 DV, 6985 HV, and 79,509 BBV; 2019: 23,867 RV, 482 DV, 8711 HV, and 97,574 BBV) (Figure 1).

With the emergence of SARS-CoV-2 and arrival in the UK in early 2020, the demand for SARS-CoV-2 testing dramatically increased from 2670 in the first quarter to 83,588 by the fourth quarter of 2020. During the peak of the pandemic in 2020 and 2021 the level of testing remained fairly constant (2020: 44,376 RV, 411 DV, 8719 HV, and 68,156 BBV; 2021: 33,962 RV, 285 DV, 8796 HV, and 78,212 BBV), with the start of 2022 remaining constant (2022 Q1: 2140 RV, 413 DV, 1919 HV, and 8188 BV).

Figure 1 shows a steep rise in RV in the first and second quarters of 2020, this was due to clinicians requesting full respiratory virus screening which included SARS-CoV-2; as the pandemic progressed new assays were released that exclusively targeted SARS-CoV-2 (Figure 2). While the laboratory capacity for SARS-CoV-2 testing increased to as high as 5390 sample per week during the peak of 2020, the laboratory was still required to provide its normal service, maintaining a relatively constant testing rate for non-SARS-CoV-2 tests with 121,662 tests and 121,255 in 2020 and 2021, respectively. During the same period the sample positivity rate also remained relatively unchanged for all groups, with the average positive percentages for RV at 3.8% (SD 3.4), 13.7% for DV (SD 3.9), 39.4% for HV (SD 2.4), and 1.5% for BBVs (SD 0.2).

## 4. Discussion

The SARS-CoV-2 pandemic placed significant pressure on the virology department to introduce new assays, molecular instrumentation, and employ more staff to meet the demands of screening hospital admissions patient as well as the broader Leicestershire population.

However, as the pandemic progressed there were other factors that increased pressure on the laboratory virology service. The increased capacity required the laboratory to move into a 24 h testing and reporting system, leading to the employment of locum staff (adding a new financial burden). Increased staffing levels and uncertainties over the main mode of viral transmission and effective use of PPE (personal protective equipment) [16,17,18,19], resulted in sporadic outbreaks of SARS-CoV-2 in the laboratory that required staff to self-isolate. Additionally, at this time, there was high turnaround of staff members as temporary staff were employed from regional locum agencies. This caused continuous retraining of new staff as they were moved between laboratories.

At the start of the pandemic, the first assay capable of detecting SARS-CoV-2 released by WHO, implemented in the laboratory in a low-throughput process [11]. The publication and implementation of this first assay during the early pandemic was clearly vital, but in practice the capacity limitations did not meet requirements. Through the first and into the second wave of the pandemic the laboratory rapidly adapted to increase testing capacity by introducing new instruments which allowed improved automation and increased capacity, which enabled the laboratory to rapidly change and adapt in response to constantly changing testing algorithms as the pandemic evolved- including the addition of rapid on-site ‘hot lab’ testing for patient triage [20]. This occurred against a background of a global exponential demand for laboratory reagents and consumables, causing major supply chain issues. The use of a range of suppliers and assays mitigated the impact of this to some extent.

Figure 1 shows a dramatic increase in BBV screening in 2019, in comparison to 2018, 2020, and 2021. This may appear as an anomaly but without data from previous years it is difficult to determine whether this peak is a genuine increase above normal. It would be expected that due to lockdown and social mixing restrictions that screening for BBV would be reduced during the pandemic, but our data revealed that the seasonal peak still occurred.

A dramatic spike in testing for RV is also observed in during the first two quarters of 2020 (Figure 1). As cases of SARS-CoV-2 increased there were more requests for full respiratory virus screening. This was in part due to changes in the pathology testing request system used by the hospital as SARS-CoV-2 was included in the full respiratory screening panel during the early phases before being separated into a single request. Additionally, during the early phase of the pandemic there was confusion over the precise clinical presentation of SARS-CoV-2 compared to other RVs, leading to more requests for full RVs screening in a ‘catch-all’ screening approach. Then, it was decided that to reduce the overall testing burden, hospital guidelines were modified to allow clinicians to only perform full screening in specific groups (patients on the intensive care unit, immunocompromised patients, and paediatrics). Later in the pandemic, the testing algorithm continued to evolve as specific requesting for symptomatic and asymptomatic screening was made available, as well as rapid screening for surgery patients.

Interestingly, Figure 1 shows that for HV and DV, screening either remained constant or continued to follow seasonal peaks and troughs. As pandemic restrictions were gradually lifted in 2021, the number of requests for DV steadily increased.

During the pandemic several new variants emerged that had a significant impact on the Governments response, including more regional lockdowns, new laws for enforcing isolation and wearing of face masks. The new variants could still be detected by the majority of existing PCR assays [21]. However, our test data showed that these did not influence the testing demand or capabilities (Figure 2).

There have been relatively few studies on the impact of SARS-CoV-2 testing on other diagnostic tests. One Spanish study, from a more general microbiology laboratory, covering the early part of the pandemic (2019–2020), demonstrated a massive rise in the number of virology tests, without breaking these down into those for specific viruses [22]. Our study covering a longer period before and during the pandemic (2018–2022), focusing on just virology testing, is therefore a useful complementary study to this one.

This report has some limitations in that it is a single-site study that only focuses on diagnostic virology testing, and essentially reports on our operational laboratory service experience within the context of our local testing protocols and clinical service needs. The impact of the COVID-19 pandemic on local diagnostic services may differ elsewhere.

In summary, our Leicester-based Virology Department, like all Virology Departments in NHS laboratories across the UK, eventually managed to meet the unprecedented demands of the SARS-CoV-2 pandemic, while mostly maintaining their levels of non-SARS-CoV-2 PCR testing to support our clinical teams treating patients in other specialties.

## Figures and Tables

**Figure 1 viruses-14-02233-f001:**
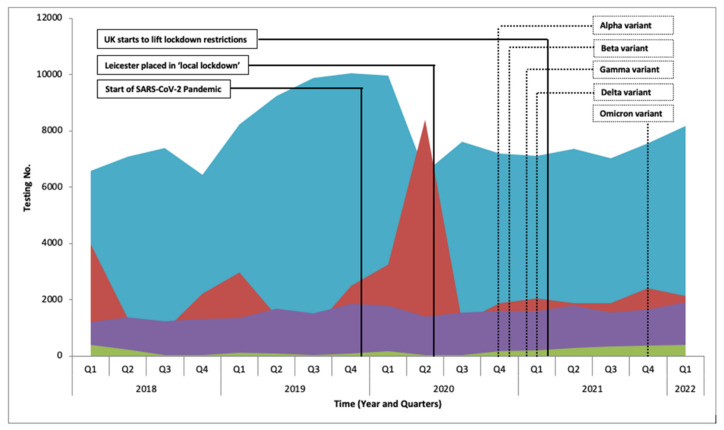
Routine Viral Screening (excluding SARS-CoV-2). Viruses are grouped into four groups; blood borne viruses (blue), respiratory viruses (red), herpes viruses (purple), diarrhoeal viruses (green).

**Figure 2 viruses-14-02233-f002:**
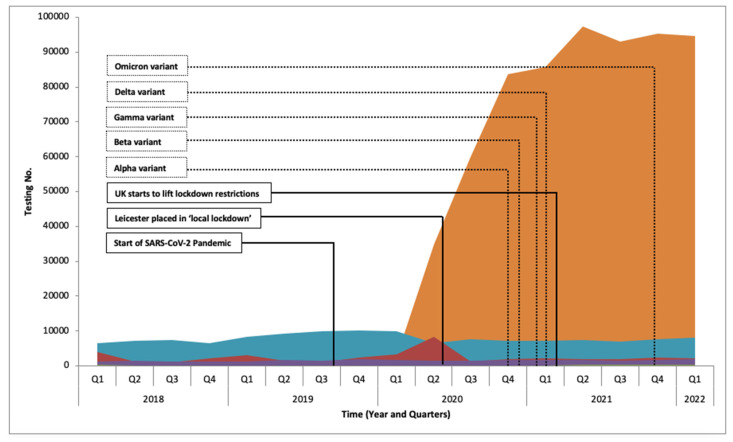
Routine Viral Screening (including SARS-CoV-2). Viruses are grouped into five groups; SARS-CoV-2 (orange), blood borne viruses (blue), respiratory viruses (red), herpes viruses (purple), diarrhoeal viruses (green).

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
