# Peer review of "Performing under Pressure: Insights into the Diagnostic Testing Burden at a UK National Health Service Clinical Virology Laboratory during the SARS-CoV-2 Pandemic"

_viruses, 2022, doi:10.3390/v14102233_

Round 1

Reviewer 1 Report

It is noteworthy that, despite what is described in most institutions, the authors state that the rest of the laboratory's workload was not affected by the introduction of this new request. It could be commented and tried to find and explanation.

At the first paragraph of Results the total number of samples processed at 2018 and 2019 are shown. At this point it would be nice to have same figures for 2020 and 2021 (and maybe the first half of 2022) to have a quick view of the huge impact of the pandemics at the laboratory.

Figures have to be revised. At least in my copy, there are some solid and dotted lines emerging from the x-axis that go to nowhere. The colors at the figures and those described at their legends do not match.

At the second paragraph of Discussion, line 5, please define “PPE”.

There is a typographic error at the last paragraph of Discussion, fourth line: “tha t” instead of “that”.

I recommend the authors to consult and include the following paper with similar messages to theirs: Catalan et al. The challenge of COVID-19 for a Clinical Microbiology Department. Diagn Microbiol Infect Dis.2021 Oct: 101(2):115426. doi: 0.1016/j.diagmicrobio.2021.115426. Epub 2021 May13. PMID: 34217111; PMCID: PMC8117483.

Author Response

Reviewer One: Thank you for your comments, each of which is addressed below:

It is noteworthy that, despite what is described in most institutions, the authors state that the rest of the laboratory's workload was not affected by the introduction of this new request. It could be commented and tried to find and explanation

Thank you for this comment. Within the text we do describe fluctuations that, although consistent with seasonal trends, were on average lower than expected due to lockdown restrictions. Within Figure we show that there is marked reduction in BBV screening, RV become more consistent (following the peak of 2020) as described on lines 175 to 179 and 186 to 188.

At the first paragraph of Results the total number of samples processed at 2018 and 2019 are shown. At this point it would be nice to have same figures for 2020 and 2021 (and maybe the first half of 2022) to have a quick view of the huge impact of the pandemics at the laboratory.

Thank you for the comment. We give a quick summary of the numbers of testing for each group on lines 120 to 121. We have added a section on the beginning of 2022 in the results section lines 121-122. Note also that the trends of these figures are shown in Figures 1 and 2, graphically.

At the second paragraph of Discussion, line 5, please define “PPE”.

Thank you, we have defined PPE in the text.

There is a typographic error at the last paragraph of Discussion, fourth line: “tha t” instead of “that”.

Thank you, we have corrected this typo.

I recommend the authors to consult and include the following paper with similar messages to theirs: Catalan et al. The challenge of COVID-19 for a Clinical Microbiology Department. Diagn Microbiol Infect Dis.2021 Oct: 101(2):115426. doi: 0.1016/j.diagmicrobio.2021.115426. Epub 2021 May13. PMID: 34217111; PMCID: PMC8117483.

Thank you for the recommendation of this paper, we missed it during our reading around the subject. It covers both Microbiology (bacteriology) and Virology testing, and only for 2019-2020, and does not separate out the testing for individual viruses, as we do here. Nevertheless, it shows a massive rise in Virology testing during 2020, as we do, and is complementary to our findings. We have now cited this in our revised Discussion.

Reviewer 2 Report

The manuscript with ID viruses-1939211 studied data of requests received by the Leicester-based NHS Virology laboratory from Jan 2018 to May 2022 and showed that a dramatic increase in SARS-CoV-2 screening but keep the same level of service for non-respiratory virus testing. The main results in figure 1 and 2 are not clear. The authors did not order the figures well and the colors are not corresponding to the ones in the figures and the solid lines and dashed lines are not explained well. So it is not possible to review well in the current format.

Author Response

Reviewer 2. We thank the Reviewer for his/her helpful comments. We have improved and uploaded the Figures which should hopefully answer his/her queries:

The main results in figure 1 and 2 are not clear. The authors did not order the figures well and the colors are not corresponding to the ones in the figures and the solid lines and dashed lines are not explained well. So it is not possible to review well in the current format.

Thank you and apologized for the error during uploading the figures. We have made the necessary amendments to both figures.

Reviewer 3 Report

Authors show in their brief report "Performing Under Pressure: Insights into the Diagnostic Testing Burden at a UK National Health Service Clinical Virology  Laboratory During the SARS-CoV-2 Pandemic" that the Leicester-based NHS Virology laboratory maintained blood borne-, herpes-, respiratory-, and diarrhoeal-viruses diagnosing during the COVID-19  pandemic at the same levels as the pre-pandemic levels regardless of the high burden that the SARS-CoV-2 testing imposed on the laboratories involved with virological diagnostics. The notion of a reduction in other viruses’ diagnostics during the different phases of the COVID-19 pandemic has been intuitive and speculative, thus this present work brought the realistic picture on this matter, although in a specific national health system scenario, contributing to discussions related to public health management. However, the present manuscript has minor scientific relevance in the away it is presented. If the focus would be the percentage on detection of these viruses during the so diverse phases of the pandemic in the UK, for instance, a data that authors tangentially touched upon. Exploring these percentages in more details, stratifying the data in terms of geographical location, time, and social distancing occurrence or not, would have had a greater scientific impact.

-       Figures 1 and 2 need to be better presented, the color coding in the legend does not match what is shown of the figure itself, which make it very difficult to analyze. For instance, in figure2, one cannot appreciate respiratory viruses (there is no red on the chart). Also, the amount of testing for diarrhoeal viruses in figure 1 and 2 don’t match, since figure 2 would represent the same amounts of other viruses, with only the addition of SARS-CoV-2, why is the amount diarrhoeal viruses testing so much higher?

-       The solid and dotted lines that appear in figures are not explained, what do they represent?

-       Several parts of the results are mentioned again in the discussion, maybe authors could merge the results and discussion sections.

Author Response

Reviewer Three: We thank the Reviewer for their comments - we have improved the figure and uploaded it. Note that the vertical scales on the two Figures are different - due to the addition of the SARS-CoV-2 testing. This should address the Reviewer's queries on the Figures. To address the reviewer's main query:

However, the present manuscript has minor scientific relevance in the away it is presented. If the focus would be the percentage on detection of these viruses during the so diverse phases of the pandemic in the UK, for instance, a data that authors tangentially touched upon. Exploring these percentages in more details, stratifying the data in terms of geographical location, time, and social distancing occurrence or not, would have had a greater scientific impact.

Response: We thank the Reviewer for his/her comments, but we are not clear exactly what he/she means. This report is less of a novel scientific finding, but more of an operational impact report that we believe will be of use to other diagnostic laboratories, as we all share our traumatic experiences of diagnostic testing during the COVID-19 pandemic. We have indicated on Figs 1 and 2 where the different SARS-CoV-2 variants and pandemic restrictions occurred, but not in any great detail (and levels of non-compliance were very variable in the population) as this is only meant to give some context and a timeline in which the routine non-SARS-CoV-2 diagnostic testing took place. There is only one location – Leicester, UK – so the geographical context is already defined, and of course, our experience is only related to our diagnostic laboratory in this single site population.

Round 2

Reviewer 3 Report

Authors addressed all issues/comments raised and the manuscript was improved with the new versions of figures 1 and 2. 

I would suggest that the authors incorporate their response to my comments in the discussion, in order to point out study limitations/caveats.

Author Response

The Reviewer has requested that we include some of the limitations that they noted, in the Discussion of our final version. We have now done this in the new second-to-last paragraph:

This report has some limitations in that it is a single-site study that only focuses on diagnostic virology testing, and essentially reports on our operational laboratory service experience within the context of our local testing protocols and clinical service needs. The impact of the COVID-19 pandemic on local diagnostic services may differ elsewhere.